

# Effects of shinbuto and ninjinto on prostaglandin E$_2$ production in lipopolysaccharide-treated human gingival fibroblasts

Toshiaki Ara and Norio Sogawa

Department of Pharmacology, Matsumoto Dental University, Shiojiri, Nagano, Japan

## ABSTRACT

Previously, we revealed that several kampo medicines used for patients with excess and/or medium patterns (kakkonto (TJ-1), shosaikoto (TJ-9), hangeshashinto (TJ-14), and orento (TJ-120)) reduced prostaglandin (PG)E$_2$ levels using LPS-treated human gingival fibroblasts (HGFs). Recently, we examined other kampo medicines used for patients with the deficiency pattern [bakumondoto (TJ-29), shinbuto (TJ-30), ninjinto (TJ-32), and hochuekkito (TJ-41)] and the herbs comprising shinbuto and ninjinto using the same experimental model. Shinbuto and ninjinto concentration-dependently reduced LPS-induced PGE$_2$ production by HGFs, whereas hochuekkito weakly reduced and bakumondoto did not reduce PGE$_2$ production. Shinbuto and ninjinto did not alter cyclooxygenase (COX) activity or the expression of molecules involved in the arachidonic acid cascade. Therefore, we next examined which herbs compromising shinbuto and ninjinto reduce LPS-induced PGE$_2$ production. Among these herbs, shokyo (*Zingiberis Rhizoma*) and kankyo (*Zingiberis Processum Rhizoma*) strongly and concentration-dependently decreased LPS-induced PGE$_2$ production. However, both shokyo and kankyo increased the expression of cytosolic phospholipase (cPL)A$_2$ but did not affect annexin1 or COX-2 expression. These results suggest that shokyo and kankyo suppress cPLA$_2$ activity. We demonstrated that kampo medicines suppress inflammatory responses in patients with the deficiency pattern, and in those with excess or medium patterns. Moreover, kampo medicines that contain shokyo or kankyo are considered to be effective for the treatment of inflammatory diseases.

## INTRODUCTION

Periodontal disease is an inflammatory disease of the gingiva that destroys periodontal tissues. In severe cases, alveolar bone is absorbed. In inflammatory responses and tissue degradation, prostaglandin E$_2$ (PGE$_2$), interleukin (IL)-6, and IL-8 play important roles. As PGE$_2$ has several functions in vasodilation, the enhancement of vascular permeability and pain, and osteoclastogenesis induction, PGE$_2$ participates in inflammatory responses and alveolar bone resorption in periodontal disease (*Noguchi & Ishikawa, 2007*).

Corresponding author
Toshiaki Ara, ara_t@po.mdu.ac.jp

Previously, we reported that several kampo medicines, shosaikoto (TJ-9) (*Ara et al., 2008b*), orento (TJ-120) (*Ara et al., 2010*), hangeshashinto (TJ-14) (*Nakazono et al., 2010*), and kakkonto (TJ-1) (*Kitamura, Urano & Ara, 2014*), suppress lipopolysaccharide (LPS)-induced PGE$_2$ production by human gingival fibroblasts (HGFs). Moreover, we found that shokyo, kanzo, and keihi, which are herbs contained in kakkonto, reduce PGE$_2$ production (*Ara & Sogawa, 2016*). These results suggested that these kampo medicines and herbs have anti-inflammatory effects in periodontal disease.

However, these kampo medicines are used for patients with the excess pattern or medium pattern. Kampo medicine used for those with the deficiency pattern remains to be elucidated. In the present study, we therefore examined the anti-inflammatory effects of the kampo medicines for patients with the deficiency pattern [bakumondoto (TJ-29), shinbuto (TJ-30), ninjinto (TJ-32), and hochuekkito (TJ-41)], which are used for the treatment of inflammatory diseases. Furthermore, we examined the effects on PGE$_2$ production using herbs comprising the kampo medicines that reduce PGE$_2$ production.

## MATERIALS AND METHODS

### Reagents

Kampo medicines (bakumondoto, shinbuto, ninjinto, and hochuekkito) were purchased from Tsumura & Co. (Tokyo, Japan). Powders of 8 herbs (bukuryo, bushi, kankyo, kanzo, ninjin, shakuyaku, shokyo, and sojutsu) were provided by Tsumura & Co. The ingredients in shinbuto and ninjinto formulas are shown in Tables 1 and 2. Powders of kampo medicines or herbs were suspended in Dulbecco's modified Eagle's medium (D-MEM; Sigma, St. Louis, MO, USA) containing 10% heat-inactivated fetal calf serum, 100 units/ml penicillin, and 100 mg/ml streptomycin (culture medium), and were rotated at 4 °C overnight. Then, the suspensions were centrifuged and the supernatants were filtrated through a 0.45 μm-pore membrane. Lipopolysaccharide (LPS) from *Porphyromonas gingivalis* 381 was provided by Professor Nobuhiro Hanada (School of Dental Medicine, Tsurumi University, Japan). Arachidonic acid was purchased from Cayman Chemical (Ann Arbor, MI). Other reagents were purchased from Nacalai tesque (Kyoto, Japan).

### Cells

HGFs were prepared as described previously (*Nakazono et al., 2010*). In brief, HGFs were prepared from free gingiva during the extraction of an impacted tooth with the informed consent of the subjects who consulted Matsumoto Dental University Hospital. The free gingival tissues were cut into pieces and seeded onto 24-well plates (AGC Techno Glass Co., Chiba, Japan). HGFs were maintained in culture medium at 37 °C in a humidified atmosphere of 5% CO$_2$. For passage, HGFs were trypsinized, suspended, and plated into new cultures in a 1:3 dilution ratio. HGFs were used between the 10th to 15th passages in the assays. This study was approved by the Ethical Committee of Matsumoto Dental University (No. 0063).

### Measurement of cell viability

The numbers of cells were measured using WST-8 (Cell Counting Kit-8; Dojindo, Kumamoto, Japan) according to the manufacturer's instructions. In brief, the media

**Table 1 The ingredients in the shinbuto formula.**

| Japanese name | Latin name | Amount (g) | Amount (g/g of product)[*] |
|---|---|---|---|
| bukuryo | *Poria Sclerotium* | 4.0 | 0.089 |
| shakuyaku | *Paeoniae Radix* | 3.0 | 0.067 |
| sojutsu | *Atractylodis Lanceae Rhizoma* | 3.0 | 0.067 |
| shokyo | *Zingiberis Rhizoma* | 1.5 | 0.033 |
| bushi | *Processi Aconiti Radix* | 0.5 | 0.011 |
| Total | | 12.0 | 0.267 |

**Notes.**

*7.5 g of shinbuto product contains 2.0 g of a dried extract of the mixed crude drugs.

**Table 2 The ingredients in the ninjinto formula.**

| Japanese name | Latin name | Amount (g) | Amount (g/g of product)[*] |
|---|---|---|---|
| kankyo | *Zingiberis Processum Rhizoma* | 3.0 | 0.083 |
| kanzo | *Glycyrrhizae Radix* | 3.0 | 0.083 |
| sojutsu | *Atractylodis Lanceae Rhizoma* | 3.0 | 0.083 |
| ninjin | *Ginseng Radix* | 3.0 | 0.083 |
| Total | | 12.0 | 0.333 |

**Notes.**

*7.5 g of ninjinto product contains 2.5 g of a dried extract of the mixed crude drugs.

were removed by aspiration and the cells were treated with a 100-$\mu$l mixture of WST-8 with culture medium for 2 h at 37 °C in $CO_2$ incubator. Optical density was measured (measured wavelength at 450 nm and reference wavelength at 655 nm) using an iMark microplate reader (Bio-Rad, Hercules, CA, USA), and the mean background value was subtracted from each value. Data is represented as means ± S.D. ($n = 4$).

## Measurement of prostaglandin $E_2$ ($PGE_2$), interleukin (IL)-6, and IL-8

HGFs were seeded in 96-well plates (10,000 cells/well) and incubated in serum-containing medium at 37 °C overnight. Then, the cells were treated with varying concentrations of each kampo medicine (0, 0.01, 0.1, or 1 mg/ml) or each herb (0, 10, 30, or 100 $\mu$g/ml) in the absence or presence of LPS (10 ng/ml) for 24 h (200 $\mu$l per well) in triplicate or quadruplicate for each sample. After the culture supernatants were collected, viable cell numbers were measured using WST-8 as described above.

The concentrations of $PGE_2$, IL-6, and IL-8 in the culture supernatants were measured by enzyme-linked immunosorbent assay (ELISA) according to the manufacturer's instructions ($PGE_2$), Cayman Chemical; IL-6 and IL-8, Thermo Fisher Scientific Inc., Camarillo, MA, USA), and were adjusted by the number of viable cells. Data are represented as pg or ng per 10,000 cells (mean ± S.D.).

## Measurement of cyclooxygenase (COX)-2 activity

COX-2 activity was evaluated as reported previously (*Wilborn et al., 1995*) with slight modification. In brief, to estimate COX-2 activity, HGFs were treated with LPS and herbs for 8 h, washed, and incubated in culture medium containing exogenous arachidonic acid

(10 $\mu$M). The concentrations of PGE$_2$ in the supernatants were measured by ELISA. Data are represented as pg per 10,000 cells (mean $\pm$ S.D.).

## Preparation of cell lysates

HGFs were cultured in 60-mm dishes and treated with combinations of LPS and herbs for the indicated times. Then, cells were washed twice with Tris-buffered saline, transferred into microcentrifuge tubes, and centrifuged at 6,000 $\times$ g for 5 min at 4 °C. Supernatants were aspirated and cells were lysed on ice in lysis buffer (50 mM Tris–HCl, pH 7.4, 1% Nonidet P-40, 0.25% sodium deoxycholate, 150 mM NaCl, 1 mM ethyleneglycol bis(2-aminoethylether)tetraacetic acid (EGTA), 1 mM sodium orthovanadate, 10 mM sodium fluoride, 1/100 volume of protease inhibitor cocktail (Nacalai tesque)) for 30 min at 4 °C. Samples were next centrifuged at 12,000 $\times$ g for 15 min at 4 °C, and supernatants were collected. The protein concentration was measured using a BCA Protein Assay Reagent kit (Pierce Chemical Co., Rockford, IL, USA).

## Western blotting

The samples (10 $\mu$g of protein) were fractionated in a polyacrylamide gel under reducing conditions and transferred onto a polyvinylidene difluoride (PVDF) membrane (Hybond-P; GE Healthcare, Uppsala, Sweden). The membranes were blocked with 5% ovalbumin for 1 h at room temperature and incubated with primary antibody for an additional 1 h. The membranes were further incubated with horseradish peroxidase-conjugated secondary antibodies for 1 h at room temperature. Protein bands were visualized with an ECL kit (GE Healthcare). Densitometric values of each band were calculated using ImageJ software.

Antibodies against COX-2 (sc-1745, 1:500 dilution), cytosolic PLA$_2$ (cPLA$_2$) (sc-438, 1:200 dilution), annexin1 (sc-11387, 1:1,000 dilution), and actin (sc-1616, 1:1,000 dilution), which detects a broad range of actin isoforms, were purchased from Santa Cruz Biotechnology (Santa Cruz, CA). Antibodies against extracellular signal-regulated kinase (ERK; p44/42 MAP kinase antibody, 1:1,000 dilution) and phosphorylated ERK [Phospho-p44/42 MAPK (Thr202/Tyr204) (E10) monoclonal antibody, 1:2,000 dilution] were from Cell Signaling Technology (Danvers, MA). Horseradish peroxidase-conjugated anti-goat IgG (sc-2020, 1:20,000 dilution) was from Santa Cruz, and anti-rabbit IgG (1:20,000 dilution) and anti-mouse IgG (1:20,000 dilution) were from DakoCytomation (Glostrup, Denmark).

## Statistical analysis

Differences between groups were evaluated by the two-tailed pairwise comparison test with a pooled variance, followed by correction with the Holm method (total 10 null hypotheses; five null hypotheses without kampo vs. with kampo in the absence or presence of LPS in Fig. 1, total 10 null hypotheses; three null hypotheses without kampo vs. with kampo in the absence of LPS, three null hypotheses without kampo vs. with kampo in the presence of LPS, and four null hypotheses without LPS vs. with LPS in Fig. 2). Differences between the control group and experimental groups were evaluated by a two-tailed Dunnett's test.

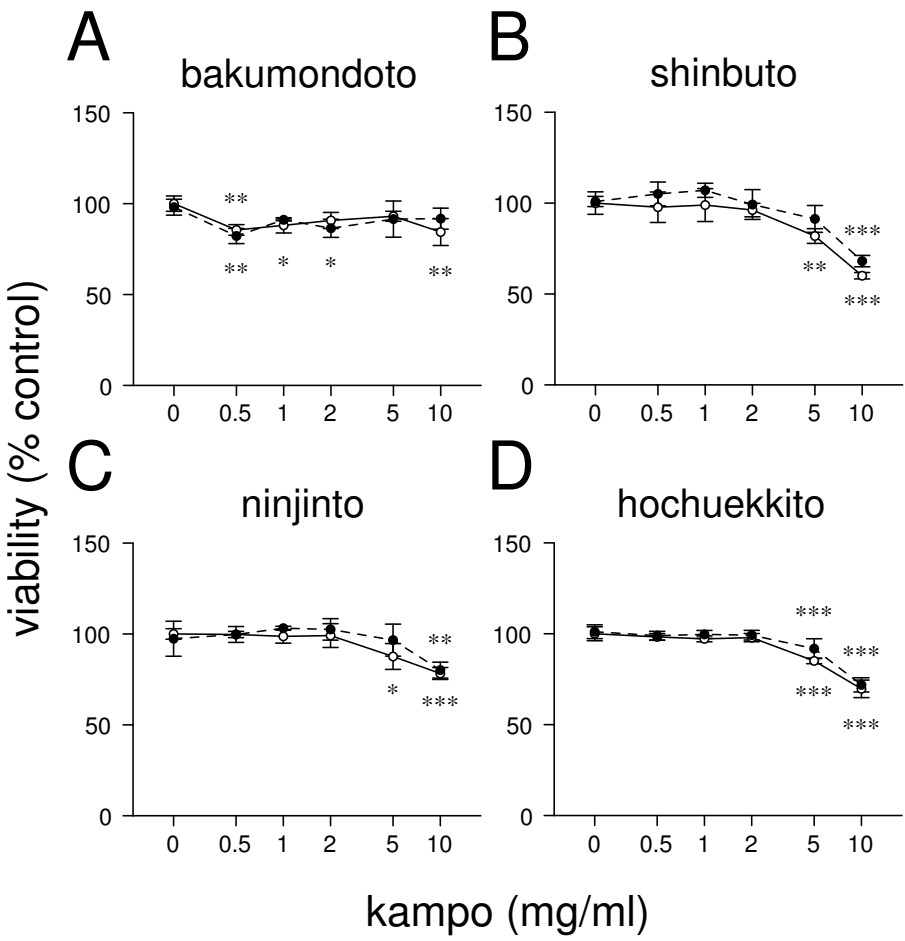

**Figure 1  Effects of kampo medicines on cytotoxicity.** HGFs were treated with combinations of LPS (0 or 10 ng/ml) and kampo medicine (0, 0.5, 1, 2, 5, or 10 mg/ml) for 24 h. Then, the numbers of viable cells were measured with WST-8. Open circles, treatment without LPS; closed circles, treatment with 10 ng/ml of LPS. *$P < 0.05$, **$P < 0.01$, ***$P < 0.001$ (without vs. with kampo medicine). $P$ values were calculated by pairwise comparisons and corrected with the Holm method (10 null hypotheses).

All computations were performed with the statistical program R (*R Core Team, 2017*). Dunnett's test was performed using the 'glht' function in the 'multcomp' package. Values with $P < 0.05$ were considered significantly different.

## RESULTS

### Effects of kampo medicines on HGFs viability

First, we examined the effects of four kampo medicines (bakumondoto, shinbuto, ninjinto, and hochuekkito) on HGFs viability. Bakumondoto did not affect the viability up to 10 mg/ml at 24 h treatment (Fig. 1A). In contrast, Shinbuto, ninjinto, and hochuekkito did not affect the viability up to 2 mg/ml, but decreased at 5 mg/ml and 10 mg/ml (Figs. 1B–1C). Therefore, up to 1 mg/ml of kampo medicines was used in further experiments because

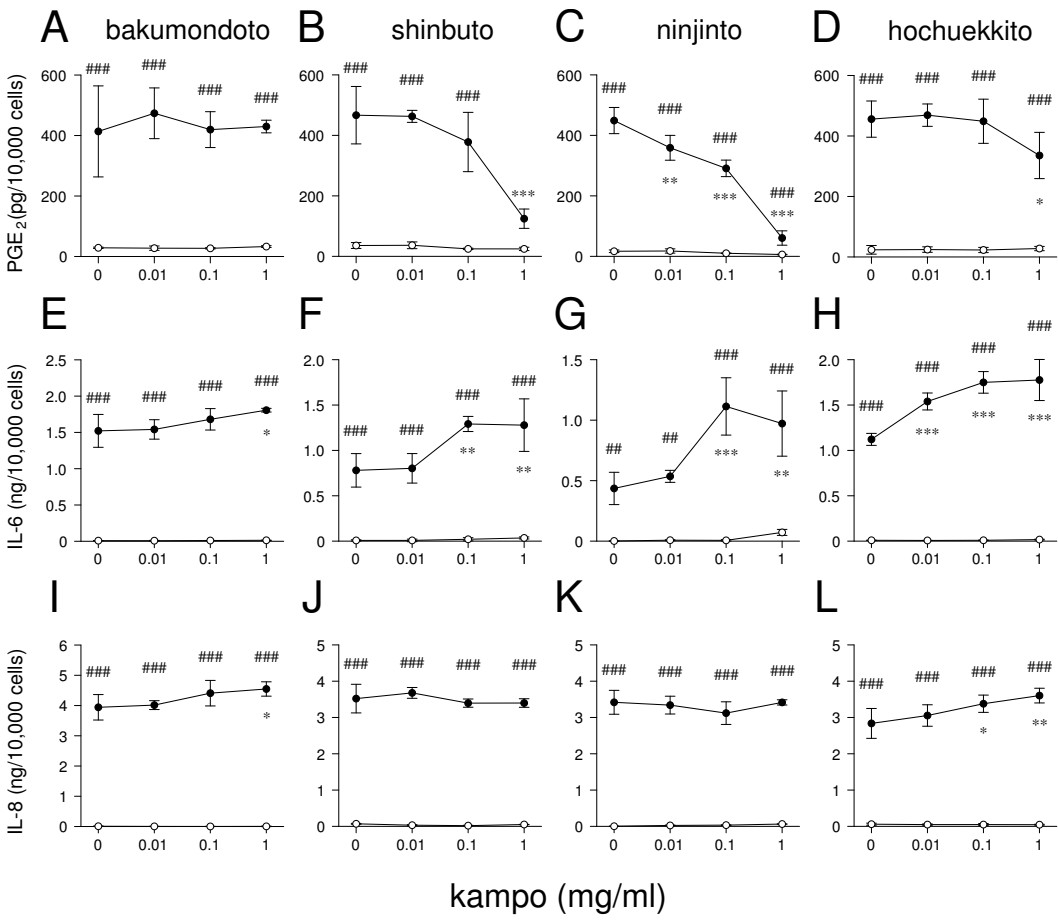

**Figure 2** **Effects of kampo medicines on PGE$_2$, IL-6, and IL-8 production.** HGFs were treated with combinations of LPS (0 or 10 ng/ml) and kampo medicine (0, 0.01, 0.1, or 1 mg/ml) for 24 h. Concentrations of PGE$_2$ (A–D), IL-6 (E–H), and IL-8 (I–L) were measured by ELISA, adjusted by cell number, and expressed as per 10,000 cells (mean ± S.D., $n = 3$). Open circles, treatment without LPS; closed circles, treatment with 10 ng/ml of LPS. $*P < 0.05$, $**P < 0.01$, $***P < 0.001$ (without vs. with kampo medicine). $^{\#}P < 0.05$, $^{\#\#}P < 0.01$, $^{\#\#\#}P < 0.001$ (without LPS vs. with LPS). $P$ values were calculated by pairwise comparisons and corrected with the Holm method (10 null hypotheses).

we used the same concentration of kampo medicines in previous studies (*Ara et al., 2008b*; *Ara et al., 2010*; *Nakazono et al., 2010*; *Kitamura, Urano & Ara, 2014*).

## Effects of kampo medicines on prostaglandin (PG)E$_2$, interleukin (IL)-6, and IL-8 production

We examined whether these kampo medicines affected the production of PGE$_2$ and inflammatory cytokines (IL-6 and IL-8) by HGFs. The concentrations of PGE$_2$, IL-6, and IL-8 were adjusted according to viable cell number. HGFs treated with 10 ng/ml of LPS produced large amounts of PGE$_2$, IL-6, and IL-8. Shinbuto and ninjinto strongly and concentration-dependently reduced LPS-induced PGE$_2$ production (Figs. 2B–2C). In contrast, bakumondoto and hochuekkito had no or little effect on PGE$_2$ production.

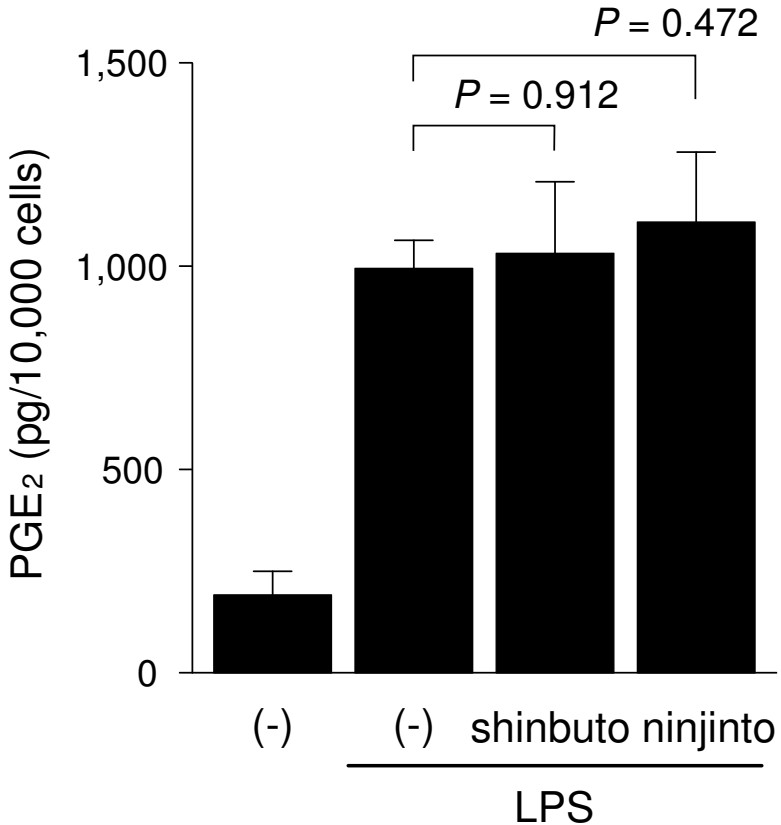

**Figure 3** **Effects of kampo medicines on COX activity.** HGFs were treated with LPS (10 ng/ml) and kampo medicine (1 mg/ml) for 8 h, washed, and then treated with 10 μM arachidonic acid for 30 min. Concentrations of $PGE_2$ were measured by ELISA, adjusted by cell number, and expressed as per 10,000 cells (mean ± S.D., $n = 4$). $P$ values by Dunnett's test are indicated.

Bakumondoto weakly, and shinbuto, ninjinto, and hochuekkito strongly increased LPS-induced IL-6 production (Figs. 2E–2H). Bakumondoto and hochuekkito weakly increased LPS-induced IL-8 production, but shinbuto and ninjinto did not affect IL-8 production (Figs. 2I–2L).

From these results, we selected two kampo medicines, shinbuto and ninjinto, which decreased $PGE_2$ production and used them in the following experiments.

### Effects of shinbuto and ninjinto on the arachidonic acid cascade

To clarify the mechanism of how shinbuto and ninjinto reduced LPS-induced $PGE_2$ production more directly, we examined the effects of these two kampo medicines on the arachidonic acid cascade. First, we examined the effects of shinbuto and ninjinto on COX activity. In order to bypass $PLA_2$, we added exogenous arachidonic acid to HGFs treated with LPS alone or LPS plus kampo medicine (shinbuto or ninjinto). Then, we measured the $PGE_2$ level produced by COX. However, shinbuto and ninjinto did not affect LPS-induced $PGE_2$ production (Fig. 3).

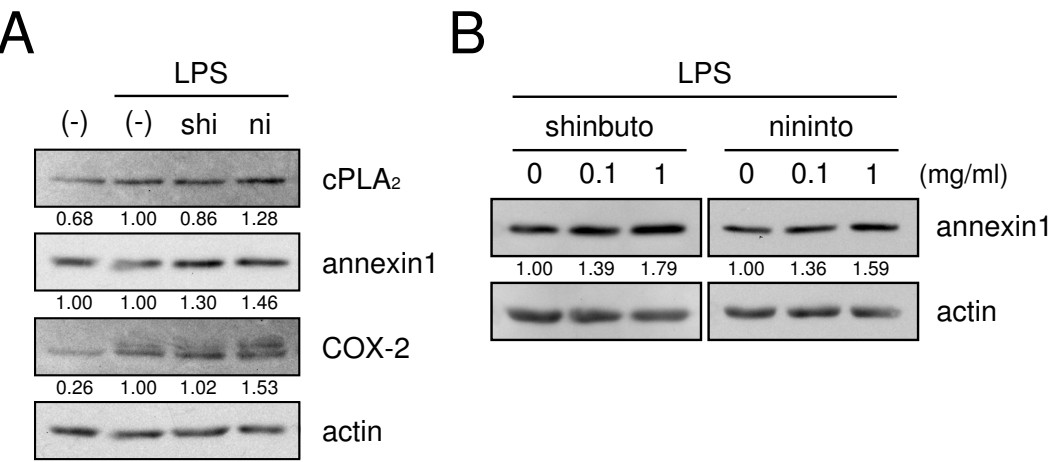

**Figure 4** **Effects of kampo medicines on cPLA$_2$, annexin1, and COX-2 expression.** HGFs were treated with a combination of LPS (0 or 10 ng/ml) and kampo medicines (0 or 1 mg/ml) for 8 h, and protein levels were examined by Western blotting. The band densities were normalized against LPS alone and actin, and indicated below each band. shi, shinbuto; ni, ninjinto.

Next, we examined whether shinbuto and ninjinto affected the expression of molecules in the arachidonic acid cascade. cPLA$_2$, which is the most upstream enzyme in the arachidonic acid cascade, releases arachidonic acid from plasma membranes. Shinbuto slightly reduced cPLA$_2$ expression and ninjinto slightly increased cPLA$_2$ expression (Fig. 4A). COX-2 was weakly expressed in the absence of LPS, and the treatment with LPS alone increased COX-2 expression. However, shokyo did not alter but kankyo slightly increased LPS-induced COX-2 expression (Fig. 4). Annexin1 (also named lipocortin1) is produced by glucocorticoids and inhibits cPLA$_2$ activity (*Gupta et al., 1984*; *Wallner et al., 1986*). Shinbuto and ninjinto slightly increased annexin1 expression (Fig. 4A) in a concentration-dependent manner (Fig. 4B).

Lastly, we evaluated the effects of shinbuto and ninjinto on ERK phosphorylation. cPLA$_2$ is directly phosphorylated and activated by phosphorylated ERK (*Lin et al., 1993*; *Gijón et al., 1999*). Therefore, we examined whether shinbuto and ninjinto suppressed LPS-induced ERK phosphorylation. LPS treatment enhanced ERK phosphorylation at 0.5 h and its phosphorylation was attenuated. However, 1 mg/ml of shinbuto or ninjinto did not affect LPS-induced ERK phosphorylation (Fig. 5).

## Effects of herbs on PGE$_2$ production and molecular expression in the arachidonic acid cascade

We examined whether herbs which comprising shinbuto and ninjinto affected LPS-induced PGE$_2$, IL-6 and IL-8 production by HGFs. When HGFs cells were treated with 10 ng/ml of LPS, HGFs cells produced large amounts of PGE$_2$. Bukuryo increased LPS-induced PGE$_2$ production. Shokyo, kankyo and kanzo strongly and significantly reduced LPS-induced PGE$_2$ production (Fig. 6A). Moreover, shokyo and kankyo decreased PGE$_2$ production in a concentration-dependent manner (Figs. 6D–6E). Other herbs had little or no effect on

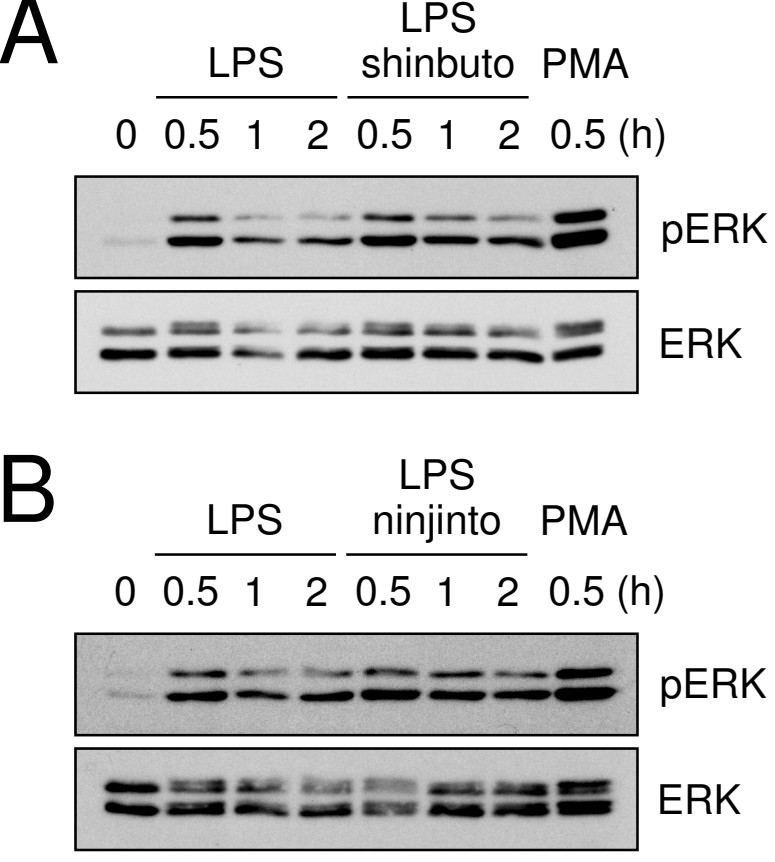

**Figure 5** **Effects of kampo medicines on LPS-induced ERK phosphorylation.** HGFs were untreated (0 h), treated with LPS (10 ng/ml), or treated with both LPS and kampo medicines (1 mg/ml) for 0.5, 1, or 2 h. PMA was used as a positive control. Western blotting was performed using anti-phosphorylated ERK or anti-ERK antibodies. pERK, phosphorylated ERK. The upper band indicates ERK1 (p44 MAPK) and lower band ERK2 (p42 MAPK).

PGE$_2$ production. Bukuryo increased LPS-induced IL-6 and IL-8 production, and kankyo increased IL-8 production (Figs. 6B–6C). Kanzo reduced IL-6 production (Fig. 6B).

We then examined whether shokyo and kankyo affected the expression of molecules in the arachidonic acid cascade. Both shokyo and kankyo increased the expression of cPLA$_2$ but did not affect annexin1 or COX-2 expression (Fig. 7).

## DISCUSSION

In our previous studies, we reported the importance of HGFs in the study of periodontal disease (*Kamemoto et al., 2009*; *Ara et al., 2010*; *Nakazono et al., 2010*; *Ara et al., 2012*; *Kitamura, Urano & Ara, 2014*; *Ara & Sogawa, 2016*), because HGFs are the most prominent cells in periodontal tissue. Moreover, LPS-treated HGFs produce inflammatory chemical mediators, such as PGE$_2$ and inflammatory cytokines such as IL-6 and IL-8 (*Sismey-Durrant & Hopps, 1991*; *Bartold & Haynes, 1991*; *Tamura et al., 1992*). Moreover, HGFs continue to produce PGE$_2$ (*Ara et al., 2008a*), IL-6, and IL-8 (*Ara et al., 2009*) in the presence of LPS.

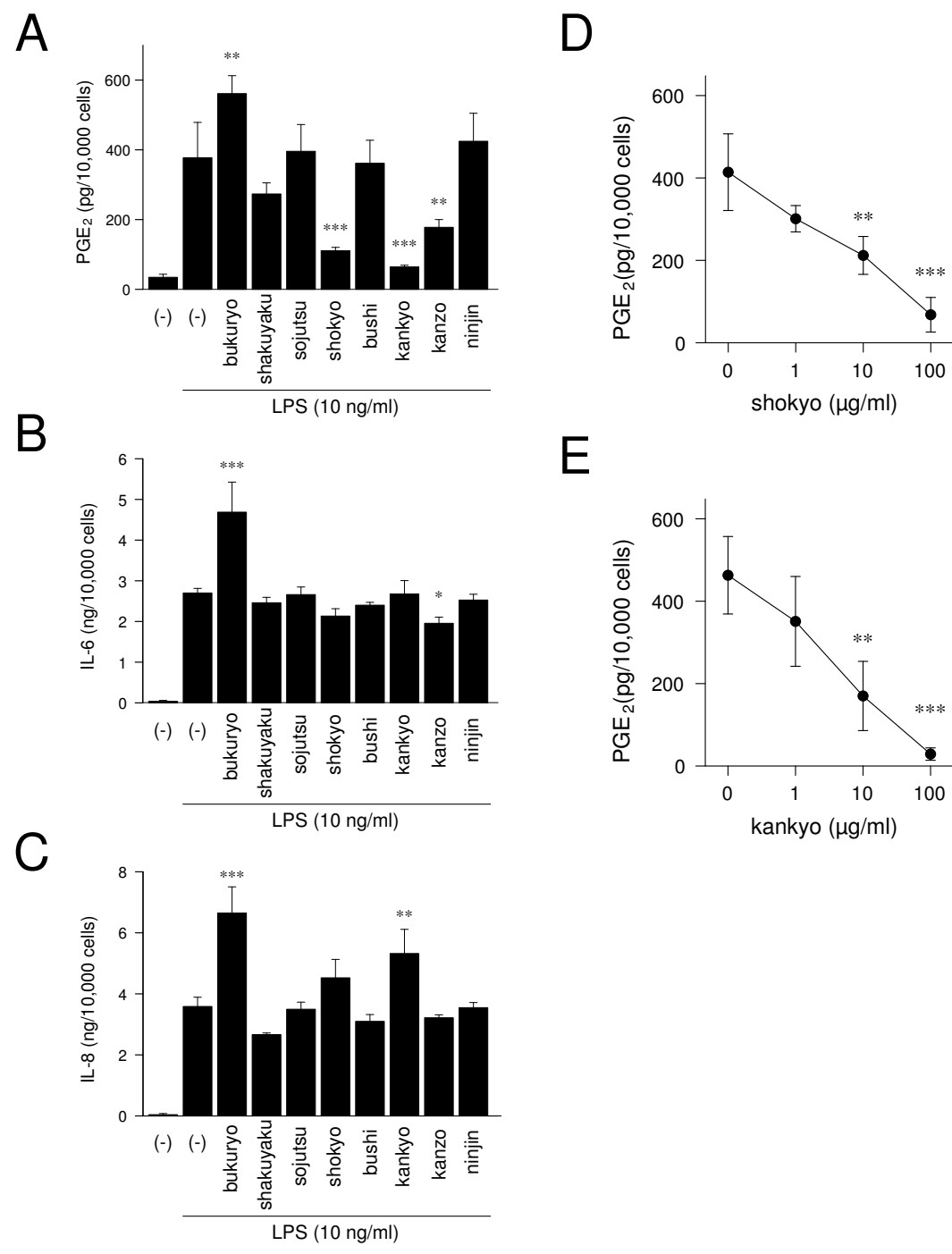

**Figure 6 Effects of herbs on LPS-induced PGE$_2$, IL-6, and IL-8 production.** (A–C) HGFs were treated with combinations of LPS (0 or 10 ng/ml) and each herb (100 μg/ml) for 24 h. Concentrations of PGE$_2$ (A), IL-6 (B), and IL-8 (C) were measured by ELISA, adjusted by cell number, and expressed as per 10,000 cells (mean ± S.D., $n = 3$). (D–E) HGFs were treated with combinations of LPS (10 ng/ml) and herbs (0, 1, 10, or 100 μg/ml) for 24 h. Concentrations of PGE$_2$ were measured by ELISA, adjusted by cell number, and expressed as per 10,000 cells (mean ± S.D., $n = 3$). *$P < 0.05$, **$P < 0.01$, ***$P < 0.001$ (LPS alone vs. LPS plus herb, Dunnett's test).

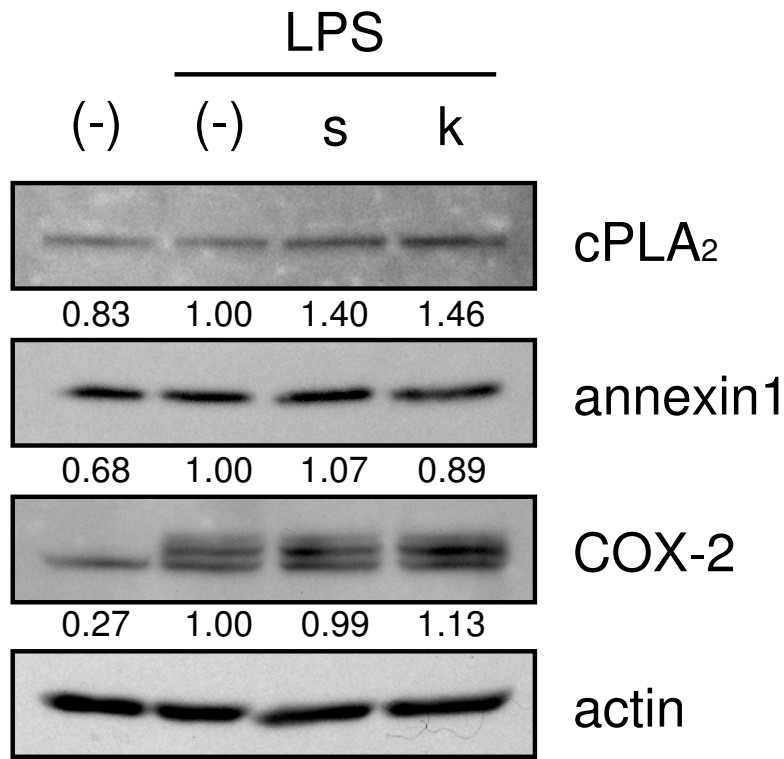

**Figure 7** **Effects of shokyo and kankyo on cPLA$_2$, annexin1, and COX-2 expression.** HGFs were treated with a combination of LPS (0 or 10 ng/ml) and herbs (1 mg/ml) for 8 h, and protein levels were examined by Western blotting. The band densities were normalized against LPS alone and actin, and indicated below each band. s, shokyo; k, kankyo.

Therefore, the large amount of chemical mediators and cytokines derived from HGFs may be contained in periodontal tissues. From these findings, we believe that examining the effects of drugs on HGFs is needed in the study of periodontal disease.

In the present study, we examined the effects of kampo medicines on LPS-induced PGE$_2$, IL-6, and IL-8 production by HGFs in patients with the deficiency pattern. Shinbuto and ninjinto dose-dependently reduceed LPS-induced PGE$_2$ production (Figs. 2B–2C), similar with shosaikoto, hangeshashinto, orento, and kakkonto (*Ara et al., 2008b*; *Nakazono et al., 2010*; *Ara et al., 2010*; *Kitamura, Urano & Ara, 2014*). However, shinbuto and ninjinto increased LPS-induced IL-6 and IL-8 production (Figs. 2F–2G, 2J–2K). In general, acid non-steroidal anti-inflammatory drugs (NSAIDs) exhibit anti-inflammatory effects by suppressing PGE$_2$ production even though they do not affect IL-6 or IL-8 production. Therefore, our results suggest that shinbuto and ninjinto have anti-inflammatory effects in periodontal disease similar with acid NSAIDs.

In the experiments at the herb level, shokyo (*Zingiberis Rhizoma*), kankyo (*Zingiberis Processum Rhizoma*), and kanzo (*Glycyrrhizae Radix*) reduced PGE$_2$ production (Fig. 6). Shokyo is contained in shinbuto (Table 1), and kankyo and kanzo are contained in ninjinto (Table 2). Shokyo is the powdered rhizome of ginger (*Zingiber offinale* Roscoe), whereas, kankyo is the steamed and powdered rhizome of ginger. Many reports demonstrated

that ginger has anti-inflammatory effects in human (*Afzal et al., 2001*; *Lakhan, Ford & Tepper, 2015*), animal (*Thomson et al., 2002*; *Aimbire et al., 2007*; *El-Abhar, Hammad & Gawad, 2008*), and *in vitro* models (*Ara & Sogawa, 2016*; *Podlogar & Verspohl, 2012*). Shokyo contains gingerols such as 6-, 8-, and 10-gingerols. With prolonged storage or heat-treatment of ginger, gingerols are converted to shogaols, which are the dehydrated form of the gingerols (*Afzal et al., 2001*). Therefore, kankyo contains the largest amount of shogaols.

Recently, we found that shokyo suppressed LPS-induced $PGE_2$ production by HGFs and that shokyo may suppress $PLA_2$ activity (*Ara & Sogawa, 2016*). In the present study, we examined the effects of kankyo in comparison with shokyo. Shokyo and kankyo increased $cPLA_2$ expression but did not alter annexin1 expression (Fig. 7). Moreover, we revealed that shinbuto and ninjinto, which contain shokyo and kankyo respectively, did not alter $PGE_2$ production when arachidonic acid was added to bypass the upstream pathway (Fig. 3). These data suggest that shokyo and kankyo did not affect the downstream pathway of arachidonic acid, which includes COX-2 and PGE synthase. In addition, shinbuto and ninjinto did not affect ERK phosphorylation (Fig. 5). From our findings described above, we were unable to explain the mechanism of the reduction in $PGE_2$ production. As gingerols in ginger are reported to inhibit both calcium-independent $PLA_2$ ($iPLA_2$) and $cPLA_2$ activities (*Nievergelt et al., 2011*), shokyo and kankyo are suggested to inhibit $PLA_2$ as discussed in the previous study (*Ara & Sogawa, 2016*). Previously, we reported that $cPLA_2$ is the main isoform in HGFs (*Ara & Sogawa, 2016*) among the subtypes such as $cPLA_2$, $iPLA_2$, and secretory $PLA_2$ ($sPLA_2$) (*Burke & Dennis, 2009*). Therefore, shokyo and kankyo may mainly inhibit $cPLA_2$ activity in HGFs. We found that orento decreases LPS-induced $PGE_2$ production via the suppression of ERK phosphorylation (*Ara et al., 2010*). However, orento may also reduce LPS-induced $PGE_2$ production by inhibition of $cPLA_2$ activity because orento contains kankyo.

We demonstrated that shokyo and kankyo concentration-dependently reduced LPS-induced $PGE_2$ production (Fig. 6A), and that the effects of kankyo are slightly stronger than those of shokyo (Figs. 6D–6E). In previous study, 6- and 8-gingerols were found to not inhibit $cPLA_2$ activity, but 10-gingerol and 6-, 8-, and 10-shogaols did (*Nievergelt et al., 2011*). Therefore, the difference in these effects on $PGE_2$ production between shokyo and kankyo may be due to the amount of shogaols in these herbs.

We demonstrated that shinbuto and ninjinto slightly increased annexin1 expression (Fig. 4). However, the involvement of annexin1 in the reduction in $PGE_2$ production is unlikely. Shokyo and kankyo did not alter annexin1 expression (Fig. 7). All 4 herbs other than shokyo in shinbuto did not reduce $PGE_2$ production, but rather, bukuryo increased $PGE_2$ production (Fig. 6A). Similarly, kanzo in ninjinto increased annexin1 expression in HGFs, and kanzo also inhibited COX activity (*Ara & Sogawa, 2016*). The 2 residual herbs other than kankyo and kanzo did not reduce $PGE_2$ production (Fig. 6A). Therefore, the increased annexin1 expression did not contribute to decreased $PGE_2$ production.

At the herb level, we were unable to clarify which herbs affect cytokine production. Bukuryo in shinbuto increased LPS-induced IL-6 and IL-8 production (Figs. 6B–6C). Therefore, this effect of shinbuto on increased IL-6 production may be due to bukuryo.

However, shinbuto did not alter IL-8 production even though it contains bukuryo. Moreover, although ninjinto increased LPS-induced IL-6 production, kanzo reduced IL-6 production, and the other three herbs, kankyo, sojutsu, and ninjin, did not alter IL-6 production. Similarly, although ninjinto did not alter IL-8 production, kankyo increased IL-8 production. Therefore, the effects of herbs on IL-6 and IL-8 production are considered to not be due to a single herb but to the combination of herbs.

Both the expression of COX-2, and the production of IL-6 and IL-8 are widely known to be regulated by NF-$\kappa$B. Ginger and its components gingerol and shogaol have been reported to suppress NF-$\kappa$B activation, and to reduce COX-2 expression and the production of IL-6 and IL-8. For example, ginger suppressed NF-$\kappa$B activation in ovarian cancer cells (*Rhode et al., 2007*), and 6-gingerol suppressed NF-$\kappa$B activation in mouse macrophage RAW264.7 cells (*Pan et al., 2008*), TPA-treated mouse skin *in vivo* (*Kim et al., 2005*), and in intestinal epithelial cells (*Saha et al., 2016*). Similarly, 6-shogaol suppressed NF-$\kappa$B activation in mouse macrophage RAW264.7 cells (*Pan et al., 2008*) and microglia cells (*Ha et al., 2012*). 6-Gingerol and 6-shogaol suppressed COX-2 expression in mouse macrophage RAW264.7 cells (*Pan et al., 2008*) and primary rat astrocytes (*Shim et al., 2011*). 6-Gingerol reduced the production of IL-1$\alpha$, IL-1$\beta$, IL-6, and IL-8 in intestinal epithelial cells (*Saha et al., 2016*). However, shinbuto and ninjinto, which contain shokyo and kankyo, respectively, increased LPS-induced IL-6 and IL-8 production by HGFs (Fig. 2) similar with kakkonto (*Kitamura, Urano & Ara, 2014*). Moreover, these two kampo medicines, shokyo, and kankyo did not suppress COX-2 expression (Figs. 4A and 7). These findings raised the possibility that shokyo and kankyo, their components, gingerols and shogaols, do not suppress the NF-$\kappa$B pathway in HGFs. The assumption is able to explain why shokyo and kankyo did not suppress COX-2 expression, which is also regulated by the NF-$\kappa$B pathway. Furthermore, 6-gingerol and 6-shogaol had no effect on LPS-induced IL-8 production in human bronchial epithelial cells (*Podlogar & Verspohl, 2012*). Therefore, the effects of gingerols and shogaols may be different among cell types.

## CONCLUSION

We demonstrated that shinbuto and ninjinto reduced LPS-induced PGE$_2$ production by HGFs. Moreover, shokyo and kankyo, which are included in these kampo medicines respectively, concentration-dependently reduced LPS-induced PGE$_2$ production. However, shokyo and kankyo did not alter the expression of the molecules in the arachidonic acid cascade, suggesting that shokyo and kankyo inhibit cPLA$_2$ activity. Therefore, the kampo medicines that contain shokyo or kankyo may have the ability to reduce PGE$_2$ production. We found that the kampo medicines used for patients with the deficiency pattern also have anti-inflammatory effects in those with the excess pattern or medium pattern. We expect kampo medicines to be used for improving inflammatory diseases, such as periodontal disease and stomatitis, in patients with any pattern.

## ACKNOWLEDGEMENTS

We thank Prof. Nobuo Yoshinari (Department of Periodontology, Matsumoto Dental University) for HGFs preparation. We also thank Prof. Nobuyuki Udagawa (Department of Biochemistry, Matsumoto Dental University) and Prof. Naoyuki Takahashi (Institute for Oral Science, Matsumoto Dental University) for their advice on our work.

### Funding

The study was supported by funding from JSPS KAKENHI Grant Number JP16H05144, the Nagano Society for the Promotion of Science, and a Scientific Research Special Grant from Matsumoto Dental University. The funders had no role in study design, data collection and analysis, decision to publish, or preparation of the manuscript.

### Grant Disclosures

The following grant information was disclosed by the authors:
JSPS KAKENHI Grant: JP16H05144.
Matsumoto Dental University.

### Competing Interests

The authors declare there are no competing interests.

### Author Contributions

- Toshiaki Ara conceived and designed the experiments, performed the experiments, analyzed the data, contributed reagents/materials/analysis tools, wrote the paper, prepared figures and/or tables, reviewed drafts of the paper, statistical analysis.
- Norio Sogawa conceived and designed the experiments, analyzed the data, reviewed drafts of the paper.

### Human Ethics

The following information was supplied relating to ethical approvals (i.e., approving body and any reference numbers):

This study was approved by the Ethical Committee of Matsumoto Dental University (No. 0063).

### Data Availability

The raw data has been provided as Supplemental Files.

### Supplemental Information

Supplemental information for this article can be found online at http://dx.doi.org/10.7717/peerj.4120#supplemental-information.

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
