# Peer review of "Effects of shinbuto and ninjinto on prostaglandin E2 production in lipopolysaccharide-treated human gingival fibroblasts"

_PeerJ, doi:10.7717/peerj.4120_

## Round 0.1 · original submission · Major Revisions

In this form the manuscript raises some questions to solve as indicated by the reviewers.

Reviewer 1 ·

Basic reporting

no comment

Experimental design

The methods are appropriate and the experiments are well performed.

Validity of the findings

no comment

Additional comments

In this manuscript, the author described the effect of shinbuto and ninjinto on prostaglandin E2 production and potential mechanism. The manuscript has potential interest to the field. The methods are appropriate and the experiments are well performed. Understanding more about the effect of kampo drugs is an essential step for medication usage. The manuscript have potential interest in the field. The methods are appropriate and the experiments are well performed. The manuscript is also acceptable to me except a few questions.

1. Line 182-185. I think “Shokyo and kankyo” here should be “shinbuto and ninjinto”.
2. Figure2 legend title is wrong. cPLA, annexin1 and cox-2 is not mentioned in this figure.
3. For the westernblot data, it is better if the statistical data is provided. Especially for fig.7. I think shoky and kankyo have effect for cPLA expression.

Reviewer 2 ·

Basic reporting

The authors investigate the anti-inflammatory effect of two kampo medicines (shinbuto and ninjinto) on human gingival fibroblasts (HGF) derived from patients affected by periodontal disease with deficiency pattern.
They show these two compounds able to decrease PGE2 production while increasing IL-6 and not affecting IL-8 levels. They exclude the modulatory effect on PGE2 effect as a consequence of COX-2 inhibition, impact on ERK activation or modulation of expression of factors cPLA2, ammexin1, and COX-2. Finally, they tested the effect on PGE2 of single herbal components composing the two formulations and identified shokyo and kankyo as inhibitors of PGE2 production without affecting the above mention factors.
The authors conclude about a potential impact of the two campo medicines on PLA2 activity.

The ms is clear; however, english proofreading would be recommended due to the presence of several mistakes.
Relevant literature is adequately provided.
Figures are clear.

Experimental design

The experimental procedures are clear as reported in Materials and Methods with ethical approval indicated regarding the use of specimens from patients .

The aims of the present investigation are clearly indicated in abstract as well as in the introduction section.

Validity of the findings

In its present form the ms, raises some questions to solve.

The two kampo medicines decrease PGE2 levels while increasing IL-6 and not affecting IL-8 production. The authors show the ability of single components shokyo and kankyo to similarly affect PGE2 levels while not affecting the expression of PLA-related factors.

1)
It would be interesting to verify if shokyo and kankyo are able to produce the same pattern of modulation of interleukin-6/8 as shinbuto and ninjinto. This phenomena might correspond to a kind of combinational effect of different components contained in shinbuto and ninjinto.

2)
The authors hypothesize the impact of shinbuto and ninjinto on cPLA2 activity. It would be interesting to provide this additional experiment to make robust their hypothesis.

---

## Round 0.2 · accepted · Accept

All reviewer remarks have been considered by the authors.

Reviewer 2 ·

Basic reporting

Authors considered all remarks and addressed the points raised.

Experimental design

The experimental procedures are clear as reported in Materials and Methods with ethical approval indicated regarding the use of specimens from patients.
Original files for consent and ethics have been provided (in their original language).

Validity of the findings

The ms has been implemented with additional experiments as suggested or strategies anyway have been considered and tempted.

Additional comments

The ms has been implemented in its experimental part.
Language has been revised.